# Reconciling North Atlantic climate modes: Revised monthly indices for the East Atlantic and the Scandinavian patterns beyond the 20th century

Laia Comas-Bru[1,2], Armand Hernández[3*]

[1] UCD School of Earth Sciences. University College Dublin. Belfield. Dublin 4. Ireland.
[2] Centre for Past Climate Change and School of Archaeology, Geography & Environmental Sciences, Reading University, Whiteknights, Reading, RG6 6AH, UK
[3] Institute of Earth Sciences Jaume Almera, ICTJA, CSIC, 08028 Barcelona, Spain

*Correspondence to*: Armand Hernández (ahernandez@ictja.csic.es)

**Abstract.** Climate variability in the North Atlantic sector is commonly ascribed to the North Atlantic Oscillation. However, recent studies have shown that taking into account the second and third mode of variability (namely the East Atlantic – EA – and the Scandinavian – SCA – patterns) greatly improves our understanding of their controlling mechanisms, as well as their impact on climate. The most commonly used EA and SCA indices span the period from 1950 to present which is too short, for example, to calibrate palaeoclimate records or assess their variability over multi-decadal scales. To tackle this, here, we create new EOF-based monthly EA and SCA indices covering the period from 1851 to present; and compare them with their equivalent instrumental indices. We also review and discuss the value of these new records and provide insights into the reasons why different sources of data may give slightly different time-series. Furthermore, we demonstrate that using these patterns to explain climate variability beyond the winter season needs to be done carefully due to their non-stationary behaviour. The datasets are available at https://doi.org/10.1594/PANGAEA.892769.

## 1 Introduction

The spatial structure of regional climate variability follows recurrent patterns often referred to as modes of climate variability or teleconnections, which provide a simplified description of the climate system (Trenberth and Jones, 2007). For example, a considerable fraction of inter-annual climate variability in the Northern Hemisphere is often ascribed to the North Atlantic Oscillation (NAO), which represents the principal mode of winter climate variability across much of the North Atlantic sector (Hurrell, 1995; Wanner et al., 2001; Hurrell and Deser, 2010) and explains c. 40% of the winter sea-level pressure (SLP) variability in the region (Pinto and Raible, 2012). However, considering other modes of variability that have historically received less attention better explains the overall regional SLP and climate variability. In particular, the East Atlantic (EA) and the Scandinavian (SCA) patterns have been demonstrated to significantly influence the winter European climate (Comas-Bru and McDermott, 2014; Hall and Hanna, 2018) as well as the sensitivity of climate variables such as temperature and precipitation to the NAO. Furthermore, the interplay of these modes exerts a strong impact on climates at different spatio-temporal scales and have important ecological and societal impacts (e.g., Jerez and Trigo, 2013; Bastos et al., 2016) as well as impacts on the availability of, for example, wind-energy resources (Zubiate et al., 2017).

In particular, the NAO consists of a N-S dipole of SLP anomalies resulting from the co-occurrence of
the Azores High and the Icelandic Low (Hurrell, 1995) and modulates the extra-tropical zonal flow. Its varying
strength is indicated by swings between positive and negative phases that produce large changes in surface air
temperature, winds, storminess and precipitation across Eurasia, North Africa, Greenland and North America
(Hurrell and Deser, 2010). The NAO is commonly described by an index calculated as the difference in
normalized SLP over Iceland and the Azores (Cropper et al., 2015; Rogers, 1984), Lisbon (Hurrell and van
Loon, 1997) or Gibraltar (Jones et al., 1997), but there are a number of robust alternatives to this classical
definition of the NAO index such as Empirical Orthogonal Function analysis (EOF; Folland et al., 2009).
The second mode of climate variability in the North Atlantic region, the EA pattern, was originally
identified in the EOF analysis of Barnston and Livezey (1987) and the exact representation of its EOF loadings
is still a matter of debate. Some authors describe the EA as a N-S dipole of anomaly centres spanning the North
Atlantic from East to West (Bastos et al. 2016; Chafik et al. 2017) while others characterise it as a well-defined
SLP monopole south of Iceland and west of Ireland, near 52.5°N, 22.5°W (Josey and Marsh, 2005; Moore and
Renfrew, 2012; Comas-Bru and McDermott, 2014; Zubiate et al., 2017). However, regardless of its exact spatial
structure, the location of its main centre of action is, in all cases, along the nodal line of the NAO; often
implying a "southward shifted NAO" with the corresponding North Atlantic storm track and jet stream also
shifted towards lower latitudes (Woollings et al., 2010). The most common methods to obtain an index for the
EA are EOF analyses (Barnston and Livezey, 1987; Comas-Bru and McDermott, 2014 Moore et al., 2013) or
Rotated Principal Component Analysis (CPC, 2012), but the SLP instrumental series from Valentia
Observatory, Ireland (51.93°N 10.23°W) has also been used in a limited number of studies (Comas-Bru et al.,
2016; Moore and Renfrew, 2012). Here we use the positive phase of the EA as a strong centre of positive SLP
anomalies offshore Ireland. This is associated with below-average surface temperatures in Southern Europe,
drier conditions over Western Europe and wetter conditions across much of Eastern Europe and the Norwegian
coast (Moore et al., 2011; Rodríguez-Puebla and Nieto, 2010).
The SCA pattern is usually defined as the third leading mode of winter SLP variability in the European
region and is equivalent to the Eurasia-1 pattern described by Barnston and Livezey (1987). It shows a vigorous
centre at 60-70°N 25-50E with some studies showing a more diffuse centre of opposite sign south of Greenland.
As far as we are aware, only EOF analyses (Comas-Bru and McDermott, 2014; Crasemann et al., 2017; Moore
et al., 2013; Hall and Hanna, 2018) and Rotated Principal Component Analysis (Bueh and Nakamura, 2007;
CPC, 2012) have been used to obtain a temporal index of the SCA. The positive phase of the SCA is related to a
higher than average pressure anomalies over Fennoscandia, Western Russia and in some cases Northern Europe,
which may lead to a blocking situation that results in winter dry conditions over the Scandinavian region,
below-average temperatures across central Russia and Western Europe and wet conditions in Southern Europe
(CPC, 2012; Bueh and Nakamura, 2007; Crasemann et al., 2017; Scherrer et al., 2005).
To the best of our knowledge, while NAO indices are available from a wide variety of sources such as
the Climate Prediction Center, CPC-NOAA (http://www.cpc.ncep.noaa.gov); the Climate Data Guide
(https://climatedataguide.ucar.edu); and the Climate Research Unit, University of East Anglia, CRU-UEA
(http://www.cru.uea.ac.uk), only the CPC-NOAA provides EA and SCA indices and, in both cases, they only
cover the period since 1950. Along the same lines, the NOAA-CIRE
(https://www.esrl.noaa.gov/psd/data/20thC_Rean/timeseries/) provides a set of climate indices created with the
20CRv2c dataset (Compo et al., 2011), but the EA and the SCA are not included. This urges scientists willing to
use a longer EA and/or SCA index to do their own EOF analyses, thereby increasing the likelihood that different
studies will use EOF-based EA and SCA indices that may be based on a different geographical area (i.e., North
Atlantic versus Northern Hemisphere), months (i.e., winter versus annual) or time-periods, while at the same
time increasing the likelihood of computational discrepancies. Therefore, making long monthly EOF-based
indices of the EA and SCA readily available will probably contribute to an increased consistency across
research studies such as those that aim at calibrating proxy-based records of past climate variability.

88       On the other hand, station-based indices have the advantage of providing continuous records that may

extend back beyond the 20th Century, when reanalysis data are more scarce (Cropper et al., 2015). However, the
main compromises of such methodology are that (i) using station-based indices implies a fixed location of the
mode's centres of action; even though non-stationarities in the geographical location of such centres, in
particular those of the NAO, have been widely demonstrated (Blade et al., 2012; Lehner et al., 2012);  (ii) the
SLP recorded by meteorological stations may not be regionally representative due to local biases (i.e. artificial
changes in their local environments; Pielke et al., 2007); and (iii) early SLP recordings may be compromised by
the use of less reliable old instrumental devices (Aguilar et al., 2003; Trewin, 2010). By contrast, while EOF-
based indices better capture the inter-annual variability in an area larger than the exact location of the centres of
action (Folland et al., 2009), they are constrained by (i) the accuracy of the reanalysis products from which they
are derived;, (ii) the non-stationarity of the EOF pattern; and (iii) the orthogonality imposed by the EOF
technique; (iv) the fact that the constructed EOFs are influenced by the region selected; and (iv) having to repeat
the analysis every time an update is required, which may change previously obtained time-series (Wang et al.,
2014; Cropper et al., 2015). It is also worth noting that the EOFs are statistical constructs and are not always
associated with climate physics (Dommenget and Latif, 2002). Here, we present a compilation of monthly
indices of the EA and the SCA based on meteorological stations and from five reanalyses products. The
instrumental series go back to 1866 and 1901, respectively, while the EOF-based series go back to 1851. To the
best of our knowledge, these are the longest EA and SCA datasets made available to the scientific community.
We also provide a comprehensive comparison of the instrumental and EOF-based indices, including their ability
to capture seasonal changes of the SLP field in the region.
**2 Data and Methods**
**2.1 Instrumental data**
A set of meteorological stations were selected according to their proximity to the EA and SCA centres
of action shown in our EOF analyses: Ireland for the EA and Norway for the SCA. Only one meteorological
station with SLP measurements in Ireland could be used in this study: Valentia Observatory. On the other hand,
five Norwegian stations with SLP data were located in the region of interest. The most suitable Norwegian
station was further selected according to three criteria: i) length of the record, ii) continuity (i.e. the least missing
data, the better) and iii) correlation with the EOF-based SCA time-series. Bergen Florida (Norway) was the
station which better fulfilled these criteria. Details of all meteorological stations are available in Table S1.
Thus, daily records from Valentia Observatory (Ireland; 01/10/1939-31/12/2016) and Bergen Florida
(Norway; 01/01/1901-31/10/2016) as well as monthly data from Valentia Observatory (January 1866 to
December 2013; Table 1) have been used to calculate the monthly series that form our instrumental indices.
Only one day (14/11/2012) and four months (December 1938; May 1872, 1873 and 1874) were missing from
the Valentia SLP data. Filling the gap in the daily time-series with its long-term average does not improve the
accuracy of the corresponding monthly mean, and so this day has been omitted in the calculations. Datasets
were tested for inhomogeneities already by their sources (Table 1). A long continuous record of monthly SLP
for Valentia was obtained by merging the monthly averages from January 1866 to December 2016 and the
computed monthly means for the period since November 1939 on the basis that the overlapping period (1939-
2013) showed a correlation $\rho > 0.99$. Hereafter, standardised monthly SLP anomalies for these stations are named
$Val_{SLP}$ and $Ber_{SLP}$.

### 2.2 Empirical Orthogonal Function (EOF) analysis

Five reanalyses datasets have been used in this study (Table 2). ERA-40 (Uppala et al., 2005) is a
conventional-input reanalysis used in many studies that require long-term atmospheric data. ERA-Interim (Dee
et al., 2011) improves ERA-40 in that it assimilates a more complete set of observations and therefore achieves
more realistic representations of the hydrologic cycle and the stratospheric circulation relative to ERA-40, as
well as it improves the consistency of the reanalysis products over time. ERA-20C (Poli et al., 2016) directly
assimilates surface pressure and surface wind observations, enabling it to extend back in time to cover the entire
20th century. 20CRv2c (Compo et al., 2011) is also a surface-input reanalysis with a different assimilation
procedure than that of ERA-20C. The main limitation of 20CRv2c is that it does not correct for biases in surface
pressure observations from ships and buoys, which results in the anomalous SLP observed for the period 1850-
1865. Finally, the NCEP/NCAR (Kalnay et al., 1996) was the first modern reanalysis of extended temporal
coverage (1948 to present) and it is still widely used. For an extensive review on the quality of these datasets,
the reader is referred to Fujiwara et al., 2017.
Empirical Orthogonal Function (EOF) analysis was performed on the above-mentioned five reanalyses
datasets of monthly SLP for a constrained Atlantic sector (100°W-40°E, 10-80°N; Table 2) using the *pca*
function of Matlab© R2018a (which is equivalent to *prin_comp* in R and the *PCA* function from the sklearn
library in Python). As in previous studies, the SLP anomalies were geographically equalized prior to the
analyses by multiplying them by the square root of the cosine of its corresponding latitude (North et al., 1982).
The percentage of variance explained by each EOF is shown in Table S2.
To maximise the representation of each pattern across seasons, and because the relative strength of the
three main modes of variability is not constant throughout the year, all EOFs have been calculated for each
three-month season (DJF, MAM, JJA and SON). Although we only used SLP fields, these patterns are also
recognisable if using different levels of the atmosphere. See Wallace and Gutzler (1981) and Cradden and
McDermott (2018) for patterns using 500-mb heights and Barnston and Livezey (1987) for 700-mb heights.
The polarities of the derived EOF time-series have been fixed to correspond to our definitions of the
EA and the SCA (see section 1), which coincide with positive centres of action offshore Ireland and over
Scandinavia, respectively (Figs. 1 and S1-S4). This is consistent with the expected climate patterns and in the
case of the EA, is compatible with the usage of SLP data from Valentia Observatory (Ireland) as an instrumental
EA index (Comas-Bru et al., 2016; Moore and Renfrew, 2012; see section 3.1).

### 2.3 Composite time-series

Monthly composite series of the NAO, EA and SCA patterns have been calculated for each 3-month
season independently. Each individual month was given the average of the available EOF-based series with a
confidence interval that corresponds to their standard deviation. The number of EOF-based series used for any
given month is provided here along with the composite series. Since the EA and the SCA do not always
correspond to the 2nd and 3rd EOF, respectively, a selection of what series to include in each composite based on
their spatial patterns was done in advance (see Table 3 for a list of individual EOFs included in each composite).
**2.4  Data analysis**
All correlations have been computed using Spearman rank coefficients (rho, $\rho$) to avoid assumptions
about normally distributed data that are inherent in some other correlation coefficients. The Spearman rank
correlation coefficient is generally expressed as Eq. (1):
$$\rho = 1 - \frac{6\sum_{i=1}^{n} d_i^2}{n(n^2-1)} \tag{1}$$
Where $n$ is the number of measurements in each of the two variables in the correlation and $d_i$ is the
difference between the ranks of the i[th] observation of the two variables.
When computing the 30-year running correlations, the significance of the correlations for each time
window was done using a Monte Carlo approach following the methodology described in Ebisuzaki (1997).
Each time window is defined from $i$ to $i+30$, where $i$ is the oldest year of overlap between the time-series.
Decadal variability of the time-series (Section 3.2.3) has been explored after filtering the time-series
with a 2nd order low-pass Butterworth filter with a cut-off frequency of 1/10 (as implemented in the *butter*
function of Matlab© R2018a).
**3 Results and discussion**
**3.1 Instrumental vs EOF-based series**
In order to identify the most suitable meteorological station to reconstruct each teleconnection index,
we first need to investigate the robustness of their spatial structures across reanalyses datasets (Figs. 1 and S1-
S4). For example, while the geographical patterns are very stable across datasets during winter (Table 3), some
discrepancies are observed during summer (JJA; see EOF2 or EOF3).
Moore and Renfrew (2012) used SLP data from Valentia Island (Ireland; Table 1) to derive an EA
station-based index and, even though this meteorological station is not located at EA centre of SLP anomalies,
the correlation coefficients between its winter values (when the mode is strongest) and EOF2 are very high
(0.7<$\rho$<0. 9; Fig. 2a; Table 4). Furthermore, our results show that when an EA pattern is identified in the
reanalysis products, the location of Valentia Observatory lies within the main area of SLP anomalies. For an
example, see the relative location of the purple dot and the yellow centre of anomalies of EOF2 in Figure 1. This
indicates the suitability of using Valentia Observatory data as a proxy of EA variability.
After an exhaustive investigation to find a long and continuous instrumental SLP dataset in
Fennoscandia as a measurement of the strength of the Scandinavian pattern, we suggest using the SLP record
from Bergen Florida (Norway; Table 1), which falls on the SCA's centre of action as shown by the pink dots in
Fig 1. This decision is further supported by the high resemblance between this meteorological dataset and the
third EOF of the winter SLP field (0.7<$\rho$<0.8; Fig. 2b; Table 4). This EOF3 corresponds to the SCA pattern
defined by Barnston and Livezey (1987) extended towards Ireland and UK and, in some cases, most of Northern
Europe (ERA-20C, ERA-40, ERA-interim and NCEP/NCAR; see Figs. 1 and S1-S4). Due to the spatial extent
of the winter's EOF3 positive centre of anomalies covering from Scandinavia to SW Ireland, $Val_{SLP}$ (purple dot
in Figure 1 and S1-S4) is unsurprisingly correlated with all winter EOF3s ($0.5 < \rho < 0.6$; Table 4).
Consistent with previous studies (e.g., Hurrell et al., 2013; Moore et al., 2013) EOF1 represents the
NAO across seasons and datasets, albeit with slight changes in the extension and/or intensity of its southern pole
(Figs. 1 and S1-S4). However, EOF2 and EOF3 are far from showing a homogeneous pattern over the course of
the four seasons and across the five reanalysis datasets.
During spring, the spatial structure of the EA (Figs. 1 and S1-S4) is recognised in EOF3. This is
consistent with the moderate to high correlations between EOF3 and $Val_{SLP}$ ($0.6 < \rho < 0.7$; Table 4). However, due
to the observed (in some cases weak) negative pole over Scandinavia, $Ber_{SLP}$ is poorly correlated to EOF3 (-
$0.4 < \rho < -0.1$; Table 4). As the spatial patterns of EOF2 show a predominant centre over the N. Atlantic Ocean (c
40°N) in all datasets, their time series are uncorrelated with our instrumental records (Figs. 1 and S1-S4, Table
4). This mode of variability is similar to the Western Atlantic (WA) pattern defined by Wallace and Gutzler
209 (1981).

Not surprisingly, the overall picture over the course of summer is a bit more complicated than in other
seasons, when most datasets are consistent. In this case, $Val_{SLP}$ shows moderate to high correlations with EOF2
($0.6 < \rho < 0.7$; Table 4) except for ERA-interim, for which the strongest correlations are observed with EOF1 and
EOF3 ($\rho = 0.6$). However, most of these EOF2s represent an extended Scandinavian pattern (Table 4) the centre
of which covers the location of Valentia Observatory, instead of the EA. A clear EA pattern is only observed for
EOF3 ERA-20C and a northwardly shifted EA pattern is found in EOF2 ERA-interim and EOF3 NCEP/NCAR
(Table 3). These discrepancies between ERA-interim and the other datasets arise because (i) EOF1 depicts a
NAO pattern with a southern pole shifted towards Northern Europe; (ii) EOF2 represents a pattern similar to a
northwardly shifted EA; and (iii) EOF3 is equivalent to the extended SCA pattern also found in winter across all
datasets (see Figs. 1 and S1-S4).
Correlations between summer $Ber_{SLP}$ and EOF3 are moderate to high only for 20CRv2c and ERA-40
($\rho > 0.6$; Table 4) because they represent the classical SCA pattern; with a centre of anomalies only over
Fennoscandia and the North Sea. However, as a result of this spatial pattern, moderate correlations are also
found with EOF2 across datasets ($0.5 < \rho < 0.7$; except ERA-interim). Regarding ERA-interim's EOF2, the weak
correlation with $Ber_{SLP}$ ($\rho = 0.3$) is due to the EA having migrated northwards. In contrast with the rest of the
seasons, and as previously noted for $Val_{SLP}$, a range of moderate to high correlations are observed between
summer EOF1 and $Ber_{SLP}$ as a result of the observed "summer NAO" pattern already defined in previous studies
(Blade et al., 2012; Folland et al., 2009).
In the case of autumn, a more coherent picture across datasets is observed: EOF1 represents a NAO
with a weak southern pole that, in some cases, migrates towards Europe; EOF2 is equivalent to the EA with a
weak negative pole over Scandinavia; and EOF3 shows a SCA pattern similar to the one obtained for the winter
months. Consequently, $Val_{SLP}$ is correlated with EOF2 ($0.6 < \rho < 0.7$) and $Ber_{SLP}$ to the EOF3 ($0.6 < \rho < 0.8$) for all
the reanalysis products. However, due to the extended SCA in EOF3, $Val_{SLP}$ is also moderately correlated to it
for all datasets except ERA-interim, where Valentia Observatory lies at the edge of the centre. In addition,
$Val_{SLP}$ is also moderately correlated with ERA-interim's EOF1 as a result of the NAO's southern pole being
shifted towards NW Europe (Fig. S3).
In summary, it has been shown that winter and autumn $Val_{SLP}$ and $Ber_{SLP}$ indices correlate with EOF2
and EOF3, respectively. In contrast, the summer EA and SCA patterns swap their order in some datasets but
good correlations are found when the geographical representation of the EOFs is taken into account. During
spring, the EA pattern is represented by EOF3 across all datasets, and EOF2 shows the WA pattern. In this case,
the SCA pattern is not reflected in any of the first three components of the EOF analysis.

**3.2 New monthly EA and SCA time-series**

**3.2.1 Monthly composites**

Each reanalysis dataset has advantages and shortcomings when it comes to its ability to reproduce the
different climate modes and, outlining objective indicators to select the reanalysis dataset that performs best is
outside of the scope of this study. Instead, since the correlations amongst datasets are very high (DJF: $\rho<0.9$;
MAM: $\rho>0.8$; JJA: $\rho>0.6$; SON: $\rho>0.9$; Table S3), we have created robust composite series of each climate
mode on the basis of their geographical representations as described in Table 3. This was done by averaging the
overlapping EOF-based time-series that display either the NAO, EA or SCA (WA for MAM). See section 2.2
for further details.
Figures 3 and 4 show the monthly time-series of $EA_{comp}/SCA_{comp}$, $Val_{SLP}/Ber_{SLP}$ and $EA_{cpc}/SCA_{cpc}$ (the
longest available records from CPC, 2012). Spearman rank coefficients between these series are in Tables 5 and
6. For winter, $Val_{SLP}$ is robustly correlated with $EA_{comp}$ ($\rho=0.8$) and moderately correlated with $SCA_{comp}$ ($\rho=0.5$;
Table 5). This results from the fact that the datasets forming $SCA_{comp}$ all show an "extended SCA" pattern
(which covers UK and Ireland, and therefore Valentia Observatory; see Figs 1 and S1-S4). On the other hand,
$Ber_{SLP}$ exhibits a very high correlation ($\rho=0.8$) with $SCA_{comp}$ and is uncorrelated with $EA_{comp}$, even though all
EA spatial patterns show a weak secondary pole of negative SLP anomalies over Scandinavia (Figs. 1 and S1-
S4). It seems therefore that only the main centre of action is reflected in the correlations (Table 5).
With regard to spring, $Val_{SLP}$ is moderately correlated with $EA_{comp}$ ($\rho=0.7$) and uncorrelated with the
$WA_{comp}$ ($\rho=0.1$). On the other hand, $Ber_{SLP}$ is uncorrelated with either $EA_{comp}$ or WA index (Table 5) because
Bergen Florida lies at the edge of the SLP dipole resulting in this station being insensitive to these climate
patterns (purple dot in Figs. 1 and S1-S4).
For summer, $Val_{SLP}$ shows a low ($\rho=0.4$) and medium-to-high ($\rho=0.5$) correlation with $EA_{comp}$ and
$SCA_{comp}$, respectively. The low correlation between $Val_{SLP}$ and $EA_{comp}$ for this season reflects the inconsistency
of the EA pattern across the different reanalysis datasets (note that the degree of correlations amongst EOFs is
the lowest in summer; Table S3). Consequently, only three datasets – ERA-20C, ERA-interim and
NCEP/NCAR – were used to construct the summer $EA_{comp}$ (Table 3) with the last two showing a clear northern
migration of its anomaly centre that leaves Valentia Observatory outside the area sensitive to this pattern (pink
dot in Figs. S3 and S4). By contrast, the observed relatively high correlation between $Val_{SLP}$ and $SCA_{comp}$ is due
to the extended SCA (Figs. 1 and S1-S4). Regarding $Ber_{SLP}$, this is poorly correlated with $EA_{comp}$ ($\rho=0.2$) and
moderately correlated with $SCA_{comp}$ ($\rho=0.6$; Table 5) as a result of the robust "extended SCA" patterns used to
create $SCA_{comp}$ (Table 3).
As far as autumn is concerned, $Val_{SLP}$ displays similar moderate correlations with $EA_{comp}$ and $SCA_{comp}$
($\rho=0.5$), again as a result of the similarity between the EA and the "extended SCA" patterns. Moreover, $Ber_{SLP}$ is
negatively correlated with $EA_{comp}$ ($\rho=-0.2$) because of the negative secondary pole of the EA (see Figs 1 and S1-
S4), and highly correlated with $SCA_{comp}$ ($\rho=0.7$).

### 3.2.2 Consistency of the correlations

To assess the temporal stability of the correlations discussed above, we have calculated 30-yr moving correlations between $EA_{comp}/SCA_{comp}$ and $Val_{SLP}/Ber_{SLP}$. As evident in Figure 5, these relationships are only stationary (and constantly significant at $\rho> 0.7$) during winter, when the two atmospheric climate modes are more robustly expressed. During spring, correlations between $EA_{comp}$ and $Val_{SLP}$ vary across a large range of values: from non-significant correlations during 1880's, early and mid-20th century (ca. 1950-1965) to moderate-to-high correlations ($\rho>0.6$) during 1930's and 1990's. By contrast, the correlations between $SCA_{comp}$ and $Ber_{SLP}$ are non-significant for almost the entire time interval (1901-2016), with only two small windows – between ca. 1925 and 1935 and around 1970 – exhibiting significant correlations ($\rho\sim0.5$). This results from the spring composite in Figure 5 representing the WA instead of the SCA. The EA correlations during summer (Fig. 5a) show the largest variability, with correlations peaking in 1940's ($\rho>0.6$) and after 1980. Non-significant correlations are found for the reminding periods. Regarding summer, $SCA_{comp}$ and $Ber_{SLP}$ are moderately correlated in the interval 1930-1980 and for a short period at the end of the 20th century. Autumn $EA_{comp}$ moderately correlates with $Val_{SLP}$ except for 1895-1920 and after ca. 1990, while $SCA_{comp}$ is only significantly correlated with $Ber_{SLP}$ in the period before ca. 1935 and after ca. 1965.

These results demonstrate that the station-based indices may be used as reference during the winter season but, beyond that, they ought to be used with caution due to the non-stationary behaviour of the EA and SCA patterns. For these non-winter seasons, almost opposite patterns of significance vs non-significance are found (i.e. $EA_{comp}$ and $Val_{SLP}$ show significant correlations when the $SCA_{comp}$ and $Ber_{SLP}$ correlations are not significant and vice versa). This may result from a displacement of their respective centres of action through time, similarly that what has been suggested for other climate modes of variability (i.e., NAO, AMO, ENSO and PDO) during these seasons for last two centuries in the North Atlantic sector (Hernández et al., 2016).

### 3.2.3 Decadal variability of new EA and SCA time-series

Figures 3 and 4 show that most variability in $EA_{comp}$ and $SCA_{comp}$ is observed at inter-annual scales but some decadal variability is also evident in Figure 6. Overall, all 10-yr filtered indices fluctuate around the zero-line with no evident trend, except for one period when both series are persistently positive: during winter at the end of the 19th century (Fig. 6a). During this season, both indices show similar trends between 1880 and 1920, when a decoupling occurs. In addition, the SCA experiences a large change of sign during the first three decades of the 20th century. Focusing on spring (Fig. 6b), we observe different patterns for both the EA and the WA with an EA absolute maximum at c. 1915 and two SCA minima at c.1930 and c.1960. The extreme absolute minima at the start of the summer $SCA_{comp}$ record (Fig. 4) seems to result from a low-pressure bias in marine records (Woodruff et al., 2005, Wallbrink et al., 2009) that has affected 20CRv2c fields such as the sea-level pressure from 1851 to c. 1865 (further information on this can be found here https://www.esrl.noaa.gov/psd/data/gridded/20thC_ReanV2c/opportunities). Since the 20CRv2c is the only reanalyses dataset covering that early period, we cannot provide an alternative. Instead, this period of low-confidence has been highlighted in all our figures with a grey band. During the rest of the period, $EA_{comp}$ and $SCA_{comp}$ alternate between similar (e.g. 1965-2000) and opposite patterns (e.g. 1910-1925), with amplitudes that gradually decrease towards present. Autumn $EA_{comp}$ and $SCA_{comp}$ alternate between in-phase (e.g. 1990-2000) and out-of-phase (e.g. 1955-1965) states.

## 3.3 Composites vs CPCs

To further check the performance of our composite series, we have compared them to the most widely used series from the CPC (CPC, 2012; Figs. 3 and 4; Table 6).

The NAO index from CPC ($NAO_{cpc}$) is moderately-to-very highly correlated with our NAO-composite across all seasons (Table 6; $0.6<\rho<0.8$). The EA index ($EA_{cpc}$) shows a moderate negative correlation with winter $EA_{comp}$ ($\rho=-0.6$) and low negative correlations with the other seasons ($\rho=-0.3$; Table 6). These negative correlations are due to the fixed polarity of the EA pattern: the main anomaly centre of our EA is positive, while that of the CPC is negative (this can be seen contrasting the spatial patterns of their teleconnection patterns – http://www.cpc.ncep.noaa.gov/data/teledoc/ea_map.shtml for the EA and http://www.cpc.ncep.noaa.gov/data/teledoc/scand_map.shtml for the SCA – and our Figures 1 and S1-S4; Comas-Bru and McDermott (2014) provide an extensive discussion on this). These negative correlations are consistent with the correlations between $EA_{cpc}$ and $Val_{SLP}$ (Table 7) as well as the running correlations discussed below. Regarding the SCA index, $SCA_{cpc}$ exhibits a low correlation with $SCA_{comp}$ for all seasons ($\rho<0.4$; note that the composite for spring is reflecting the WA pattern and hence it has not been compared with the CPC indices). The moving correlations (30-year sliding window) between the seasonal $EA_{comp}/EA_{cpc}$ (Fig. 7a) and $SCA_{comp}/SCA_{cpc}$ (Fig. 7b) are consistent with the correlations in Table 6. For winter and summer, the correlations between $EA_{comp}$ and $EA_{cpc}$ are fairly constant ($\rho<-0.5$). However, non-significant correlations are obtained for autumn during the entire time period (1950-2016) and, during spring, only the period between 1970 and 2000 is significant ($\rho<-0.4$); with the exception of few time-windows at the end of the 1980's. Regarding the temporal variability of the correlations between $SCA_{comp}$ and $SCA_{cpc}$, these are only significant ($\rho>0.4$) after 1990 for the winter season (Fig. 5b).

Overall, these results suggest that the difference in methodology between our EOFs and the one followed by the CPC, and/or the difference in the reanalysis products used is not relevant for the NAO, but it becomes critical for the EOFs that account for a smaller percentage of the total SLP variance (>30% vs 10-20%; Table S2). The low correlations observed beyond the winter season could be linked to a non-stationary behaviour of the EA and SCA resulting in migrations of their centres that are not adequately captured by our methodology and/or that employed by the CPC, or in the reanalyses products from which the indices are derived.

This is further supported by the geographical displays of seasonal $EA_{cpc}$ and $SCA_{cpc}$ (see URLs above). The $EA_{cpc}$ consists of a dipole with negative anomalies that spans from the central North Atlantic Ocean to central Europe (leaving Valentia Observatory at its margin) and positive anomalies in the middle subtropical Atlantic. According to their maps, the negative pole remains geographically fixed throughout the year only varying in intensity, whereas the positive pole varies both in strength and position, being less intense and displaced towards the centre of the subtropical Atlantic in summer. On the other hand, the $SCA_{cpc}$ is essentially a primary positive centre located over Northern Scandinavia at ~70° N (for reference, Bergen Florida station is at 60° N) with weaker negative centres over Western Europe and Russia. In this case, both poles present an almost spatial stationary behaviour with their highest intensity occurring in winter. Thus, the low correlations obtained for the CPC indices and the station-based data (Table 7) could be attributed to the distance between the meteorological stations and their centres of action.

The discrepancies observed between our composite-EOFs and those from the CPC may also be attributed to: (i) the different and shorter time period considered by CPC when performing the RPCA; (ii) the

fact that the CPC considers data from all 12 calendar months whereas the EA/SCA patterns are more distinctly
developed in wintertime; (iii) the region over which CPC computed the RPCA covers all longitudes from 20 -
90 °N, whereas we have limited our computations to the N. Atlantic region (100°W-40°E, 10°-80°N); (iv) the
non-orthogonality of the RPCA; and (v) differences related to the use of SLP or 500-mb heights and/or the
accuracy of the reanalysis datasets used.
**3.4 Climate impact of the composite EA and SCA series**
Figure 8 illustrates the monthly correlation distribution maps between our composite-series ($EA_{comp}$
and $SCA_{comp}$) versus surface air temperature and precipitation amount for the four seasons (DJF, MAM, JJA and
SON) between 1901 and 2016 using the CRU-TS.4.01 dataset (Harris et al. 2014). The strongest correlations are
found in winter, when these patterns are more prominent, and are consistent with previous studies (Moore et al,
2011; Comas-Bru and McDermott, 2014; Lim, 2015).
The only European regions for which the EA impacts on precipitation are strong and robust (i.e. on the
same direction) throughout the year are the UK and Ireland. The predominantly weak correlations observed in
other regions, far from the main centres of action, could arise from the low percentages of variability explained
by each EOF pattern (<20% for EA; Table S2). Nevertheless, consistent patterns are observed in terms of
precipitation amount across all seasons except in $EA_{comp}$/JJA, which also shows an anomalous relationship with
temperature. We interpret this to be caused by the northerly shift of the EA centre of action in JJA (i.e. between
Scotland and Iceland instead of off-shore Ireland; see Table 3 and Figures S3 and S4), that hampers its influence
on the western Mediterranean region, which in turn becomes wetter with positive EA modes. Regarding the
impact of the SCA on precipitation, a similar pattern with negative correlations in northern Europe and
predominantly positive correlations in the circum-Mediterranean region, is observed across seasons, albeit with
different strengths. We observe a strong seasonality on the impact of both climate modes on surface air
temperature. Weak correlations are found for the all seasons except JJA for the EA with non-significant
correlations across all Europe in SON. The opposite is observed for SCA, where the strongest impact on air
temperature is shown in DJF (predominantly negative) and SON (predominantly positive).
Due to the low variance explained by both climate modes, they are not expected to imprint a very
strong signal on the climate and thus the extent to which these correlations would be reflected in the absolute
precipitation and temperature values will primarily depend on the concomitant state of the NAO, the main driver
of climate variability in the region (Hurrell and van Loon, 1997; Hurrell and Deser, 2010). In addition, the
impact of these atmospheric modes on the climate is not robust throughout the year. For example, none of the
datasets used in this study showed a SCA pattern within the three leading EOFs in spring.
Individual EOFs such as the EA and the SCA are statistical constructs that do not necessarily represent
a physically independent phenomenon linked (i.e. correlated) to climate variables in a robust manner. Full
characterisation of the regional atmospheric dynamics therefore requires multiple EOFs to be taken into account
(Roundy, 2015). To thoroughly characterise the climate in the region, the impacts of the EA/SCA should be
investigated in conjunction with the NAO (Moore et al., 2011; Comas-Bru and McDermott, 2014; Hall and
Hanna, 2018) but this is outside the scope of this study. As far as we are aware, such investigation does not exist
outside the winter months.

## 4 Conclusions

This study presents a new set of indices for the second and third modes of climate variability in the North Atlantic sector ($EA_{comp}$ and $SCA_{comp}$). These indices have been constructed after identifying the main patterns of variability across five different reanalysis products and have been then compared to the two meteorological stations identified as instrumental series for the EA and the SCA pattern: Valentia Observatory (Ireland) and Bergen Florida (Norway). The high resemblance between our EOF-based indices and these instrumental SLP records during winter allows both indices to be readily updated as required. Beyond this season, however, a more complex picture arises. For example, the Scandinavian pattern is not included within the first three modes of climate variability during spring and instead, the Western Atlantic pattern as described by Wallace and Gutzler (1981) dominates SLP variability after the NAO, leaving the EA as the third pattern for this season.

Our results also suggest that the difference in methodology/reanalysis products between our composite EOF-based indices and those provided by NOAA-CPC (CPC, 2012) is not relevant for the NAO but it becomes critical for the 2nd and 3rd EOF. However, despite the differences, both sets of indices display very similar and recognisable spatio-temporal patterns at inter-annual timescales (Figs. 3 and 4).

**Data availability**

The datasets consisting of the instrumental data and the monthly composite indices of NAO, EA and SCA are available at https://doi.pangaea.de/10.1594/PANGAEA.892769.

**Author contribution:**

AH identified the meteorological stations used. LCB developed the scripts and performed the data analyses. LCB and AHH designed the calculations and carried them out. The manuscript was collaboratively written by both co-authors.

**Competing interests:**

The authors declare no competing interests.

**Acknowledgements:**

We would like to thank MetÉireann and European Climate Assessment & Dataset (ECA&D) for making publicly available the meteorological datasets of Valentia Observatory (Ireland) and Bergen Florida (Norway). Support for the Twentieth Century Reanalysis Project version 2c dataset is provided by the U.S. Department of Energy, Office of Science Biological and Environmental Research (BER), and by the National Oceanic and Atmospheric Administration Climate Program Office. We acknowledge use of ECMWF reanalysis datasets (ERA-40, ERA-20C and ERA-interim) and documentation at http://www.ecmwf.int. NCEP Reanalysis data provided by the NOAA/OAR/ESRL PSD, Boulder, Colorado, USA, from their Web site at https://www.esrl.noaa.gov/psd/. AH was supported by a Beatriu de Pinós - Marie Curie co-fund contract within the framework of the FLOODES2k (2016 BP 00023) and PaleoModes (CGL2016-75281-C2) projects.

Europe. In JJA, the southern pole is weak and predominantly shifted northwards. The same pattern is found in SON, except for 20CRv2c and ERA-20C; (ii) "EA with secondary pole" means that a negative pole over Scandinavia is evident; (iii) "Extended SCA" refers to the classic SCA with the positive pole extending towards IRL and UK; and (iv) the Western Atlantic (WA) pattern in MAM/EOF2 is a dipole with a main centre over the N. Atlantic Ocean and a second weak centre over Scandinavia (both negative). See Figures 1 and S1-S4 for the corresponding maps.

Table 4: Correlation coefficients between the three monthly EOFs for winter (DJF), spring (MAM), summer (JJA) and autumn (SON) and the corresponding monthly $Val_{SLP}$ and $Ber_{SLP}$. Note: all correlations with p-val≤0.01 except (a) 0.01<p-val≤0.05; (b) 0.05<p-val≤0.1; and (c) p-val>0.1.

Table 5: Monthly correlations of our composite indices ($EA_{comp}$ and $SCA_{comp}$) and the instrumental records ($Val_{SLP}$ and $Ber_{SLP}$). (*) Spring (MAM) pattern is that of WA. See text for details. Note: all correlations with p-val≤0.01 except (a) 0.01<p-val≤0.05; (b) 0.05<p-val≤0.1; and (c) p-val>0.1.

Table 6: Monthly correlations between the CPC indices ($NAO_{CPC}$, $EA_{CPC}$ and $SCA_{CPC}$) and our composites ($NAO_{comp}$, $EA_{comp}$ and $SCA_{comp}$. Note: all correlations with p-val≤0.01 except (a) 0.01<p-val≤0.05; (b) 0.05<p-valv0.1; and (c) p-val>0.1. The SCA only has been compared to the composites for DJF, JJA and SON because spring is showing the WA pattern (see Table 4 and Figs. 1 and S1-S4 for further details).

Table 7: Monthly correlations between the $EA_{cpc}$ and $SCA_{cpc}$ and our station-based indices ($Val_{SLP}$ and $Ber_{SLP}$). Note: all correlations with p-val≤0.01 except (a) 0.01<p-val≤0.05; (b) 0.05<p-val≤0.1; and (c) p-val>0.

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

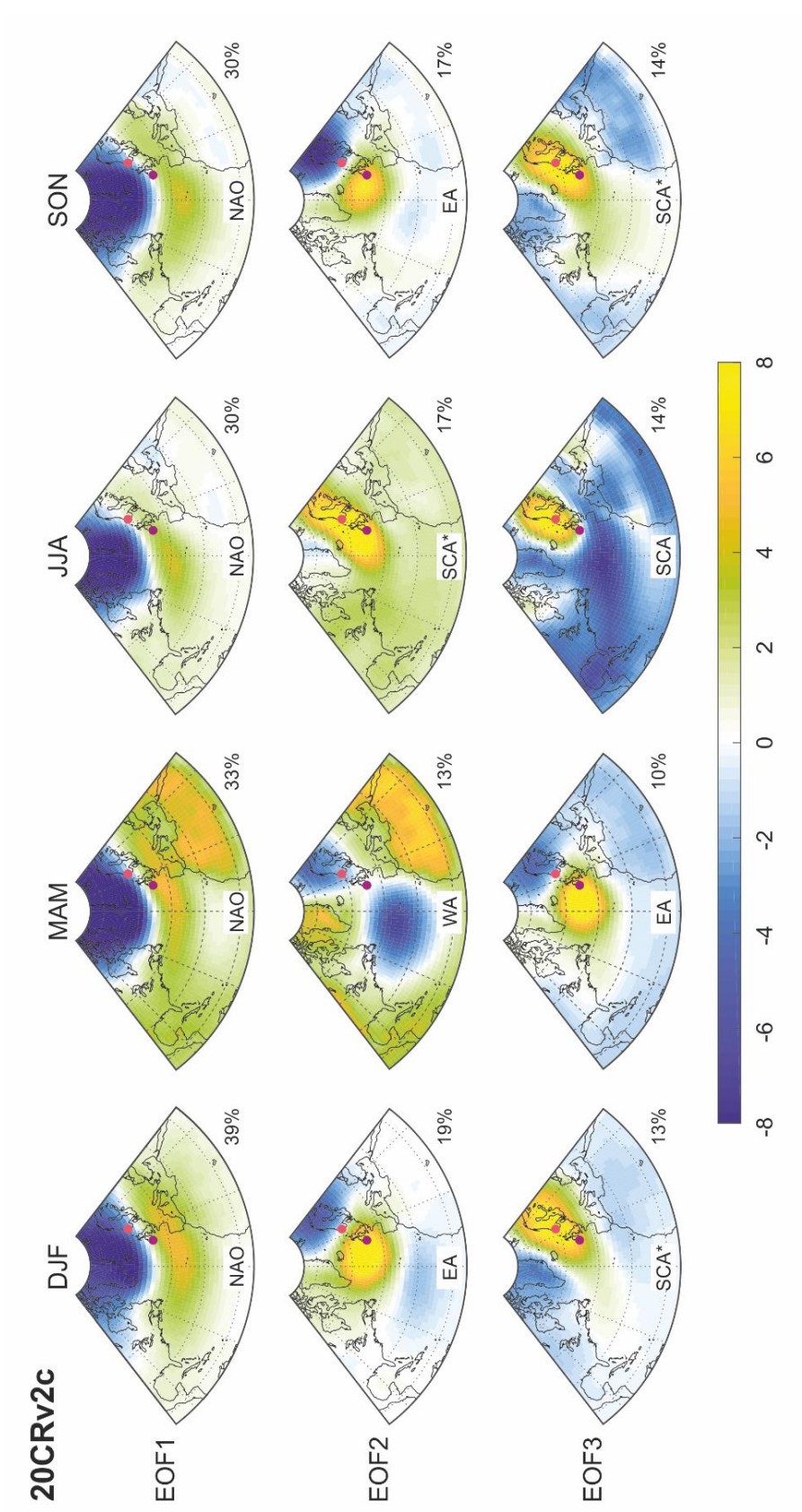

**Figure 1: EOF loadings based on monthly SLP data (20CRv2c dataset; Compo et al., 2011). Each column represents a 3-month season. The percentages at the bottom right of each map are the variability explained by the corresponding EOF (rows) at any given season (columns) as shown in Table S2. The text at the bottom of each map identifies the observed pattern. Pink (purple) dots show the location of Bergen Florida (Valentia Observatory) stations as listed in Table 1. Figures S1-S4 show the same maps for the other four reanalysis products in Table 2.**

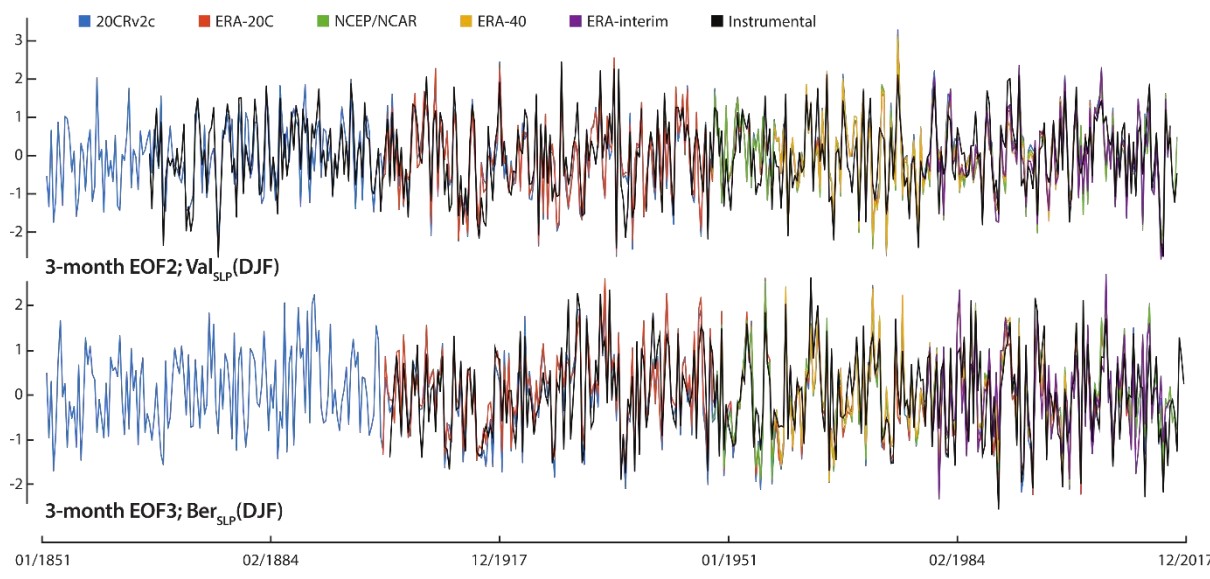


**Figure 2: Monthly (DJF) EOF time-series and their equivalent instrumental records. a) EOF2 and normalised SLP**
**data from Valentia Observatory (Val$_{SLP}$); b) same than (a) with the EOF3 and SLP data from Bergen Florida**
**(Ber$_{SLP}$). Correlation coefficients between these time-series are given in Table 4.**

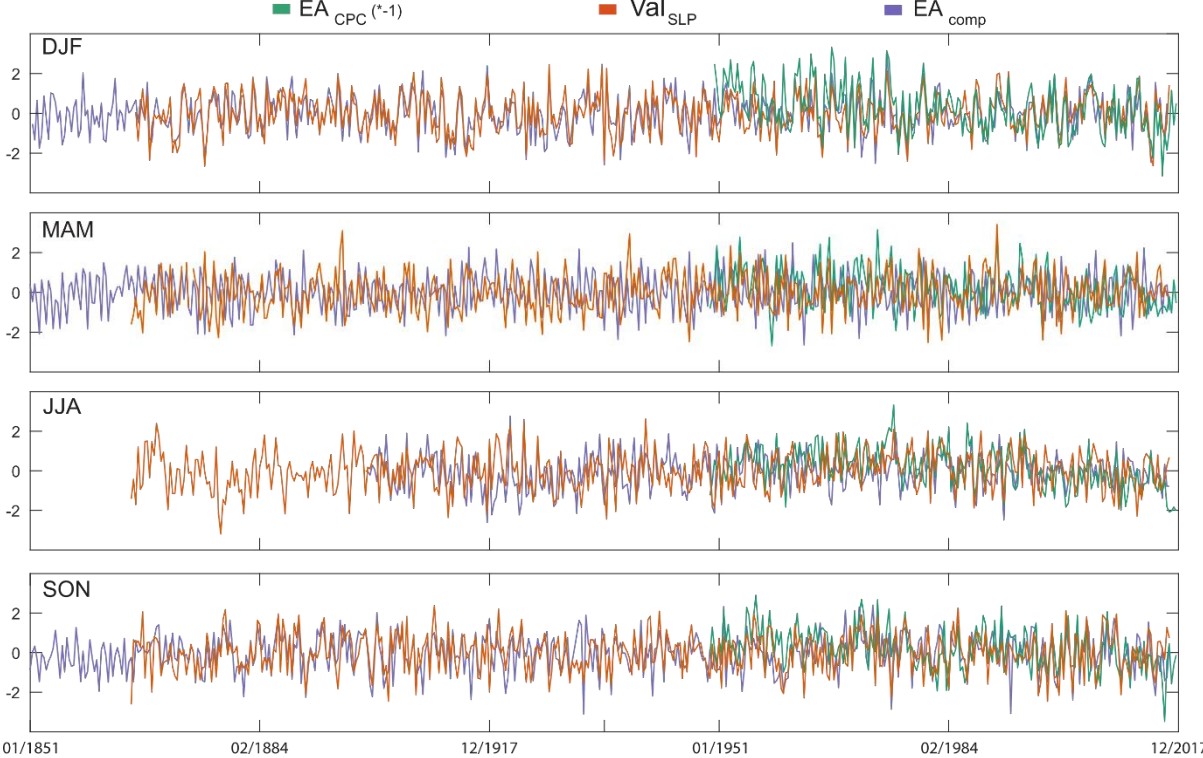


**Figure 3: Monthly series of EA$_{comp}$, the instrumental data (Val$_{SLP}$) and the EA from the CPC (EA$_{CPC}$; CPC, 2012) for each 3-months season. Note that the CPC series has been inversed for an easy visual comparison.**


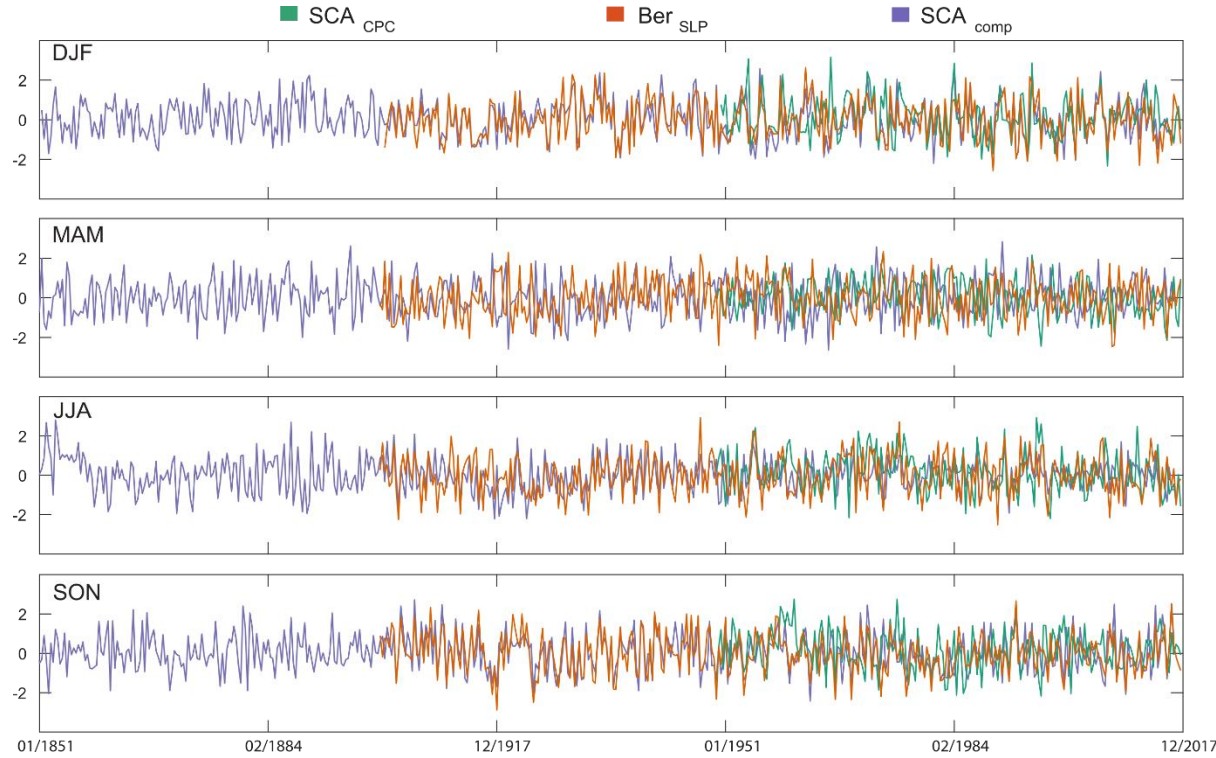


**Figure 4: Same as in Fig. 3 for SCA$_{comp}$, instrumental data (Ber$_{SLP}$) and the EA from the CPC (EA$_{CPC}$; CPC, 2012).**













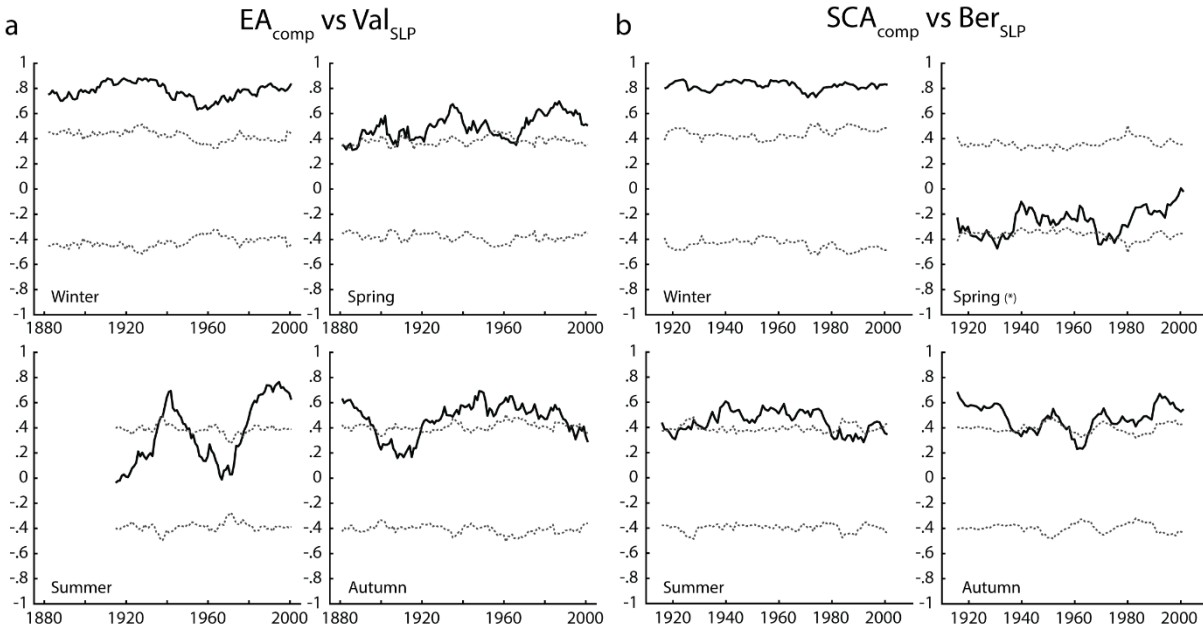



**Figure 5: Running correlations between our composite series and the instrumental records. (a) EA_comp and Val_SLP; (b) SCA_comp and Ber_SLP. The window size is 30 years and is defined from *i* to *i+30*, where *i* is the oldest month. Dashed lines indicate the 0.01 significance thresholds. Note that spring in panel (b) corresponds to the WA index instead of the SCA.**

729

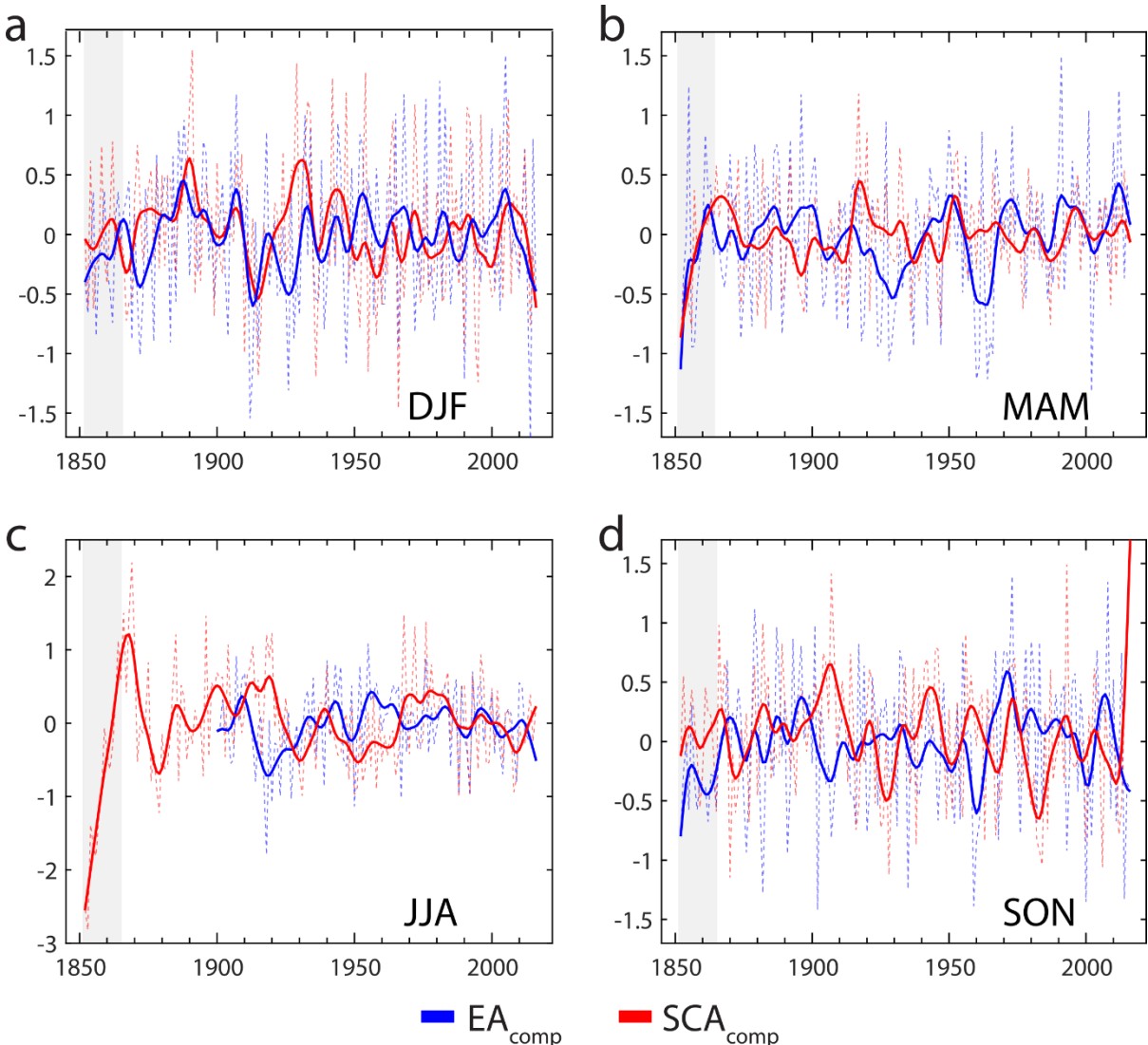

730

**Figure 6: Seasonally averaged EA$_{comp}$ (dashed blue line) and SCA$_{comp}$ (dashed red line) and decadal EA$_{comp}$ (blue solid line) and SCA$_{comp}$ (red solid line). (a) winter (DJF); (b) spring (MAM); (c) summer (JJA); (d) autumn (SON). A 10-year bandpass filter has been used to obtain the decadal series. Note that in (b) the red lines correspond to WA$_{comp}$ instead of SCA$_{comp}$. Note the different y-scale for summer indices. Grey band indicates the period of low confidence of our composite series (see methods section for details).**


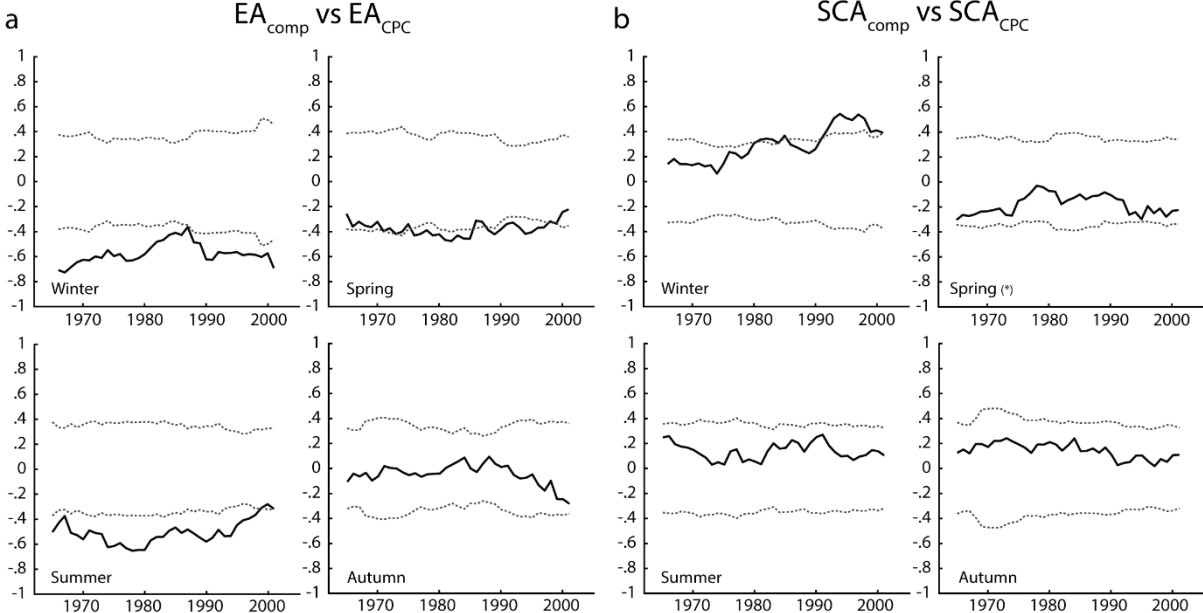


**Figure 7: Running correlations as in Fig. 5 between our composite series and the CPC indices. (a) EA$_{comp}$ and EA$_{CPC}$;**
**(b) SCA$_{comp}$ and SCA$_{CPC}$. The window size is 30 years and is defined from *i* to *i+30*, where *i* is the oldest month.**
**Dashed lines indicate the 0.01 significance thresholds.**

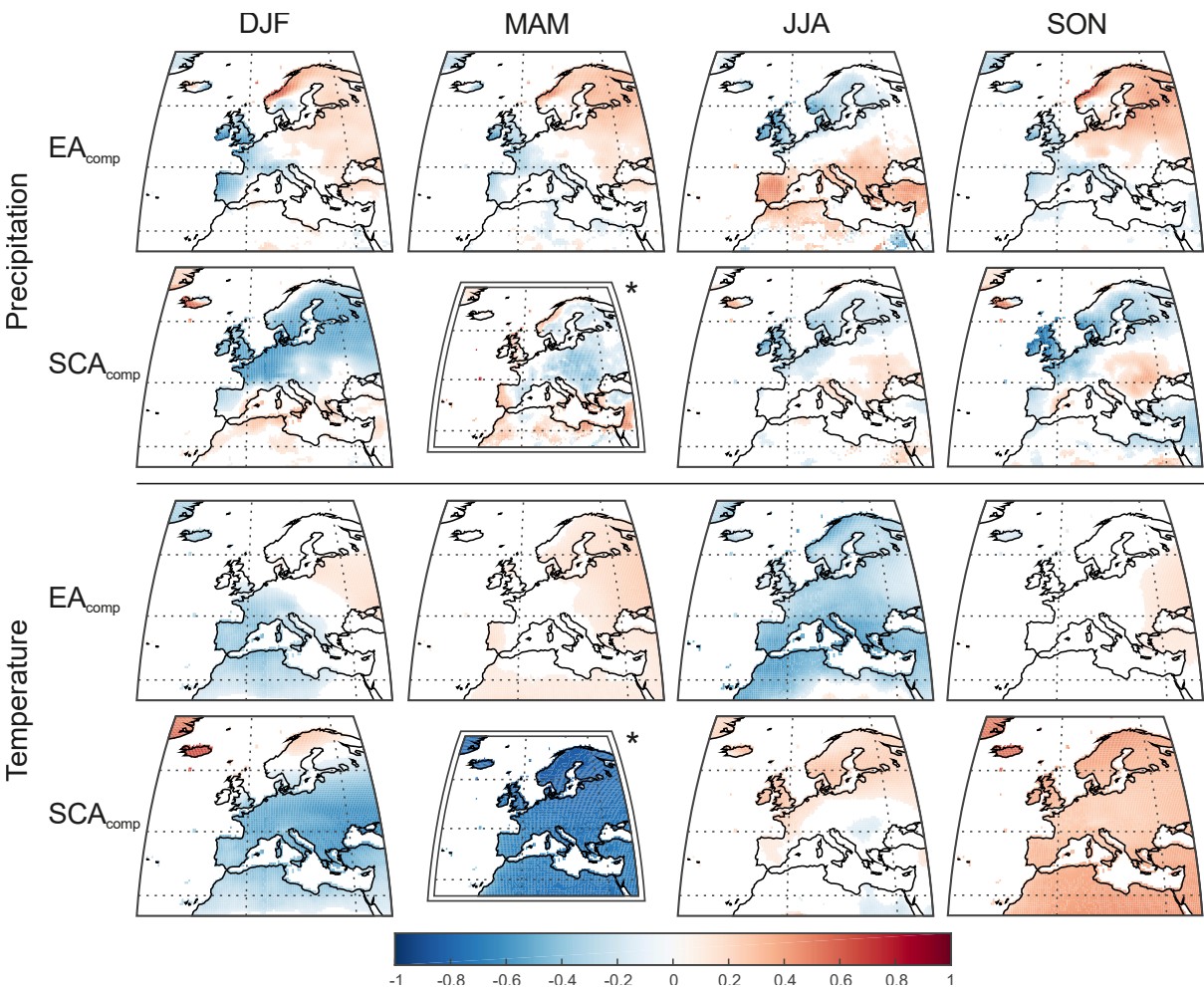

**Figure 8: Correlation distribution maps between the monthly precipitation (top) and surface air temperature**
**(bottom) and our monthly composites (EA$_{comp}$ and SCA$_{comp}$) between 1902 and 2016. Climate data from the CRU-**
**TS4.01 global climate data set (Harris et al. 2014). Positive correlations are shown in red and negative correlations**
**are shown in blue (see colour bar). Correlation coefficients are Spearman Rank coefficients. SCA$_{comp}$/MAM maps**
**(marked with an asterisk) correspond to the WA pattern.**

**Table 1. List of the meteorological stations used to construct the monthly instrumental indices. Daily data**
**downloaded from the European Climate Assessment and Dataset (ECA&D; Klein Tank et al., 2002) available at**
**www.ecad.edu (Date of last access: 6th February 2018). In April 2012 the manual station at Valentia was replaced by**
**an automatic station at the same site (Met Éireann, personal communication).**

| Station name | WMO Code | Coordinates | Altitude (m) | Time period | # missing data | Original data type | Source |
|---|---|---|---|---|---|---|---|
| Valentia Observatory | 305/22 75 | 51.94°N 10.22°W | 9 | 01/10/1939- 31/12/2016 | 1 | Daily | European Climate Assessment and Dataset (Klein Tank et al., 2002) |
| Valentia Observatory | 3953 | 51.93°N 10.25°W | 14 | 01/1866- 05/2002 | 4 | Monthly | Met Éireann |
| Bergen Florida | 265 | 60.38°N 5.33°E | 12 | 01/01/1901- 31/12/2017 | 0 | Daily | European Climate Assessment and Dataset (Klein Tank et al., 2002) |


**Table 2. Details of the reanalysis products used in this study.**

| Dataset | Description | Period | | Spatial coverage (lat x lon) | Reference |
|---|---|---|---|---|---|
| 20CRv2c | NOAA-CIRES Reanalysis dataset based on data-assimilation and surface observations of synoptic pressure | 1/1851 12/2014 | – | 2° x 2° | Compo et al. (2011) |
| NCEP/NCAR Reanalysis 1 | Reanalysis dataset based on an analysis and forecast system to perform data assimilation using past data. | 1/1948 31/2016 | – | 2.5° x 2.5° | Kalnay et al. (1996) |
| ERA-interim | ECMWF Global Reanalysis Data | 1/1979 11/2016 | – | 0.75° x 0.75° | Dee et al. (2011) |
| ERA-20C | ECMWF Reanalysis of the 20th-century using surface observations only | 1/1900 12/2010 | – | 1.125° x 1.125° | Poli et al. (2016) |
| ERA-40 | ECMWF Global Reanalysis Data | 9/1957 8/2002 | – | 1.125° x 1.125° | Uppala et al. (2005) |


**Table 3: Summary of the geographical structures of the EOF loadings across datasets (columns) and seasons (rows). Superindices indicate which EOFs are included in the composite series: [1] NAO_comp; [2] EA_comp; [3] SCA_comp; [4] WA_comp. Notes: (i) The NAO in DJF and MAM, presents a southern pole extending towards Europe. In JJA, the southern pole is weak and predominantly shifted northwards. The same pattern is found in SON, except for 20CRv2c and ERA-20C; (ii) "EA with secondary pole" means that a negative pole over Scandinavia is evident; (iii) "Extended SCA" refers to the classic SCA with the positive pole extending towards IRL and UK; (iv) the Western Atlantic (WA) pattern in MAM/EOF2 is a dipole with a main centre over the N. Atlantic Ocean and a second weak centre over Scandinavia (both negative); and (v) note that no EA pattern is observed in 20CRv2c during JJA, which results in a shorter EA_comp for this season. See Figures 1 and S1-S4 for the corresponding maps.**

| | | 20CRv2c | ERA-20C | ERA-40 | ERA-interim | NCEP/NCAR |
|---|---|---|---|---|---|---|
| DJF | EOF1 | NAO [1] | NAO [1] | NAO [1] | NAO [1] | NAO [1] |
| | EOF2 | EA with secondary pole [2] | EA with secondary pole [2] | EA with secondary pole [2] | EA with secondary pole [2] | EA with secondary pole [2] |
| | EOF3 | Extended SCA [3] | Extended SCA towards N. Europe [3] | Extended SCA towards N. Europe [3] | Extended SCA towards N. Europe [3] | Extended SCA towards N. Europe [3] |
| MAM | EOF1 | NAO [1] | NAO [1] | NAO [1] | NAO [1] | NAO [1] |
| | EOF2 | WA [4] | WA [4] | WA [4] | WA [4] | WA [4] |
| | EOF3 | EA with secondary pole [2] | EA with secondary pole [2] | EA with secondary pole [2] | EA with secondary pole [2] | EA with secondary pole [2] |
| JJA | EOF1 | NAO [1] | NAO [1] | NAO [1] | NAO [1] | NAO [1] |
| | EOF2 | Extended SCA [3] | Extended SCA [3] | Extended SCA [3] | EA (shifted to the North) [2] | Extended SCA [3] |
| | EOF3 | SCA | EA [2] | SCA | Extended SCA [3] | EA (shifted to the North) [2] |
| SON | EOF1 | NAO [1] | NAO [1] | NAO [1] | NAO [1] | NAO [1] |
| | EOF2 | EA with secondary pole [2] | EA with secondary pole [2] | EA with secondary pole [2] | EA with secondary pole [2] | EA with secondary pole [2] |
| | EOF3 | Extended SCA [3] | Extended SCA [3] | Extended SCA [3] | Extended SCA [3] | Extended SCA towards N. Atlantic ocean [3] |

**Table 4: Correlation coefficients between the three monthly EOFs for winter (DJF), spring (MAM), summer (JJA) and autumn (SON) and the corresponding monthly Val$_{SLP}$ and Ber$_{SLP}$. Note: all correlations with p-val≤0.01 except (a) 0.01<p-val≤0.05; (b) 0.05<p-val≤0.1; and (c) p-val>0.1.**

| | | 20CRv2 | | | ERA-20C | | | ERA-40 | | | ERA-interim | | | NCEP/NCAR | | |
| --- | --- | --- | --- | --- | --- | --- | --- | --- | --- | --- | --- | --- | --- | --- | --- | --- |
| | | EOF1 | EOF2 | EOF3 | EOF1 | EOF2 | EOF3 | EOF1 | EOF2 | EOF3 | EOF1 | EOF2 | EOF3 | EOF1 | EOF2 | EOF3 |
| Val$_{SLP}$ | DJF | -0.27 | 0.78 | 0.54 | 0.29 | 0.79 | 0.51 | 0.45 | 0.77 | 0.60 | 0.40[a] | 0.90 | 0.51 | 0.39 | 0.72 | 0.64 |
| | MAM | 0.25 | 0.05[c] | 0.60 | 0.22[a] | -0.20[a] | 0.58 | 0.10[c] | 0.08[c] | 0.69 | 0.40[a] | -0.28[b] | 0.71 | 0.25[a] | 0.07[c] | 0.63 |
| | JJA | 0.39 | 0.66 | 0.22 | 0.41 | 0.59 | 0.38 | 0.46 | 0.70 | 0.02[c] | 0.58 | 0.36[a] | 0.60 | 0.56 | 0.62 | 0.26[a] |
| | SON | -0.02[c] | 0.60 | 0.54 | 0.00[c] | 0.57 | 0.66 | 0.35[a] | 0.70 | 0.51 | 0.55 | 0.65 | 0.24[c] | 0.44 | 0.60 | 0.48 |
| Ber$_{SLP}$ | DJF | 0.32 | 0.05[c] | 0.80 | -0.30 | 0.10[c] | 0.80 | -0.35[a] | -0.04[c] | 0.83 | -0.60 | -0.01[c] | 0.69 | 0.35 | -0.15[c] | 0.78 |
| | MAM | -0.08[c] | -0.19[b] | -0.13[c] | -0.08[c] | -0.15[c] | -0.24[a] | -0.16[c] | -0.20[c] | -0.23[c] | 0.01[c] | -0.01[c] | 0.28[b] | 0.06[c] | -0.36 | -0.24[a] |
| | JJA | 0.33 | 0.60 | 0.60 | 0.28 | 0.52 | 0.27 | 0.50 | 0.69 | 0.61 | 0.65 | 0.35[a] | 0.28[b] | 0.52 | 0.57 | 0.05[c] |
| | SON | -0.16[b] | -0.25 | 0.82 | -0.31 | -0.21[a] | 0.79 | -0.16[c] | -0.03[c] | 0.57 | 0.06[c] | -0.00[c] | 0.66 | -0.09[c] | -0.10[c] | 0.67 |

**Table 5: Monthly correlations of our composite indices (EA$_{comp}$ and SCA$_{comp}$) and the instrumental records (Val$_{SLP}$ and Ber$_{SLP}$). (\*) Spring (MAM) pattern is that of WA. See text for details. Note: all correlations with p-val≤0.01 except [a] 0.01<p-val≤0.05; [b] 0.05<p-val≤0.1; and [c] p-val>0.1.**

| | | EA$_{comp}$ | SCA$_{comp}$ |
|---|---|---|---|
| Val$_{SLP}$ | DJF | 0.75 | 0.52 |
| | MAM | 0.65 | 0.05[c]* |
| | JJA | 0.38 | 0.48 |
| | SON | 0.55 | 0.54 |
| Ber$_{SLP}$ | DJF | 0.03[c] | 0.82 |
| | MAM | -0.10[b] | 0.08[c]* |
| | JJA | 0.23 | 0.62 |
| | SON | -0.20 | 0.71 |

**Table 6: Monthly correlations between the CPC indices (NAO$_{CPC}$, EA$_{CPC}$ and SCA$_{CPC}$) and our composites (NAO$_{comp}$, EA$_{comp}$ and SCA$_{comp}$. Note: all correlations with p-val≤0.01 except [a] 0.01<p-val≤0.05; [b] 0.05<p-valv0.1; and [c] p-val>0.1. The SCA only has been compared to the composites for DJF, JJA and SON because spring is showing the WA pattern (see Table 4 and Figs. 1 and S1-S4 for further details).**

| | | NAO$_{CPC}$ | EA$_{CPC}$ | SCA$_{CPC}$ |
|---|---|---|---|---|
| Composites | DJF | 0.81 | -0.60 | 0.41 |
| | MAM | 0.64 | -0.31 | - |
| | JJA | 0.79 | -0.31 | 0.20 |
| | SON | 0.76 | -0.39 | 0.19 |

**Table 7:** Monthly correlations between the $EA_{cpc}$ and $SCA_{cpc}$ and our station-based indices ($Val_{SLP}$ and $Ber_{SLP}$).

Note: all correlations with p-val≤0.01 except [a] 0.01<p-val≤0.05; [b] 0.05<p-val≤0.1; and [c] p-val>0.1.

| | | $EA_{CPC}$ | $SCA_{CPC}$ |
|---|---|---|---|
| **$Val_{SLP}$** | DJF | -0.58 | -0.17[a] |
| | MAM | -0.47 | -0.30 |
| | JJA | -0.36 | -0.01[c] |
| | SON | -0.54 | -0.42 |
| **$Ber_{SLP}$** | DJF | -0.26 | 0.32 |
| | MAM | -0.33 | 0.16[a] |
| | JJA | -0.26 | 0.26 |
| | SON | -0.38 | 0.24 |

