# Peer review of "Reconciling North Atlantic climate modes: Revised monthly indices for the East Atlantic and the Scandinavian patterns beyond the 20th century"

_Earth System Science Data, 2018_

## Referee Comment (RC1) · Anonymous Referee #1 · 2 Nov 2018

Reconciling North Atlantic climate modes: Revised monthly indices for the East Atlantic and the Scandinavian patterns beyond the 20th century Laia Comas-Bru, Armand Hernández Reviewer Comments General comments The paper outlines the use of climate mode indices within the North Atlantic region and identifies some limitations, particularly that the second and third modes are only available in one form from 1950 onwards. The authors construct indices from 1850, which should prove useful for studying decadal variability, and compare these with longer term station-based indices and other EOF indices from reanalysis. The paper and data are a useful contribution

to the field, but a few issues should be clarified. The paper is generally well-written and clear, with few typographical errors. Specific Comments 1. A main comment would be that I find the title, and subsequent content a bit misleading in that it purports to provide monthly indices, whereas the indices in the paper are seasonal.

2. A further methodological point is that it is not at all clear what is meant by "composites" throughout the paper, nor is it clear how these are constructed. I guess it is a combined index using different reanalyses, or do you mean combining monthly indices into seasonal indices? Exactly how these are combined should be made clear. I found this a bit confusing, but it should be stratightforward to clarify. Combination of time series from different reanalyses will involve splicing of some sort, and this should be explained clearly.

There are a number of other points listed below which should be addressed.

Page 1, Line 34: I would add the recent study by Hall and Hanna, 2018, IJOC , here, to broaden the scope of the literature. This paper also finds inconsistencies in EOFs 2 and 3 for summer. Page 2, Line 6: Other nodes are used, such as Lisbon and Gibraltar, and this should be acknowledged and referenced here Page 2, line 22: I would be more circumspect here. Although intuitively a positive EA should equate to positive SLP anomalies in line with SCA, , the CPC index is based on the reverse of this, and a number of studies take this position (Woollings et al., 2010, QJRMS; Moore et al., 2011 QJRMS; Wulff et al., 2017, GRL; Hall and Hanna, 2018, IJOC among many) so it is incorrect to promote this view of the EA as the standard one. It doesn't actually matter, the relationships are the same just inverted. It would be better to state: "Here we take the positive phase of the EA to be......" Page 2, line 29, Again it is appropriate to cite Hall and Hanna, 2018, IJOC Page 3, lines 15-20. It is also worth noting that EOFS are statistical constructs and are not always associated with climate physics (Dommenget and Latif, 2002, J. Climate). Also some acknowledgement that the constructed EOFs are influenced by the region selected. Pages 3-4 Data section. Were timeseries of station and gridded data assessed and corrected for any inhomo-

[Figure]

geneities which could arise through artificial means such as changing instrumentation, changes in density of records, etc? Page 4, line 14. It is misleading to state that the common definition of a positive EA is positive SLP anomalies, in view of the comment and references above. Change to something like "our definitions" Page 4, Line 32. How are the years of the moving windows defined, in reference to the window (start, end, centred-which is not possible with a 30 year window)? Page 6, Line 31: Are the composites monthly? They look seasonal to me. It is unclear from the text how the composites are produced. This needs to be explained clearly. This section is unclear, with confusing terminology about monthly time series when the figures show seasonal time series. Page 8. Line 6. What is the 10-year filter? Is it a simple moving average, or some sort of Gaussian filter? The caption just says "bandpass" Can you be more specific? Page 8, lines 9-10 "..until a decrease towards a minimum starts in c. 1920" It is not clear what is meant by this as from the figure the minimum appears to be reached in 1920. Page 8, lines 11-14: I don't find these descriptions particularly convincing when looking at the figure Page 8 line 15-16. I am highly sceptical about the reality of the first 20 years or so of the summer SCA figure, with its extreme maxima and minima. I think this is likely to be an artefact of data quality, See ESRL web pages https://www.esrl.noaa.gov/psd/data/gridded/20thC_ReanV2c/opportunities.html There is some evidence of this in Figure 6 panel b) as well. Technical Corrections Page 1 line 31: remove comma after "attention" Page 5 line 29: should it be DJF: p>0.9 ?

---

## Referee Comment (RC2) · Anonymous Referee #2 · 12 Nov 2018

The authors consider the second and third modes of SLP variability in the North Atlantic region and their influence on climate variables. In particular, they highlight the lack of instrumental indices and availability of long-term indices, key to study decadal climate variability. They calculate a new set of indices both based on reanalysis and instrumental data, dating back to 1850. The paper and the data are a very useful contribution to the field of climate variability and large-scale atmospheric pattern research. The text is clearly explained and overall well written. There are a few issues that I think should be considered:

[Figure]

General comments

1. Since the controlling mechanisms of the EA and the SCA are mentioned in the abstract, I would like to see a more careful description of the influence of these two patterns on climate variables and their physical meaning. Moreover, given the discrepancies across datasets and the lower percentage of variance explained by the patterns outside of the winter months, discussing their relevance would be appropriate.

2. The main contribution of this paper is, in my opinion, the introduction of the new, instrumental indices that date back longer than previously available ones and that have been proven consistent across different datasets. The correlation of the winter SCA with the Bergen station data is remarkably good. For this reason, I think it would be interesting to know more about the instrumental data, and mention which other stations were considered and why they were discarded.

Specific comments

Page 1, Line 25: "The spatial structure of climate changes..." I would suggest talking about regional climate variability rather than 'climate changes', given the data span no longer than two hundred years.

Page 2, Line 21: A strong centre of positive SLP anomalies is said here to be associated with above-average temperature and wetter conditions in Northern Europe. Please review this. Is it possible that those effects correspond to the positive phase of the reverse EA index, as used in other publications? (i.e., Moore et al., 2011)

Page 3, Line 18: The orthogonality imposed by the EOF technique should be one of the constraints listed

Page 5, Line 6: Looking at Table 3, no discrepancies are observed across datasets for EOF 3 during MAM months

Page 5, Line 21: "Because of this spatial pattern, ..." I suggest rephrasing the sentence since the meaning is not entirely clear to me

[Figure]

Page 6, Line 27: I think it is erroneous to assume that the datasets have to capture the climate modes, since the variability is in the SLP data themselves. Comparing different reanalysis datasets is out of the scope of this paper, but an indication of known quality issues that might account for the differences would be suitable

Figure 2: many data are shown on the same plot, so that it is a bit difficult to visually recognise the agreement between the series. Also, could the x axis show dates rather than number of months?

Figure 6: I suggest including DJF, MAM, JJA and SON instead of a, b, c and d on the top left corner of each panel, both for clarity and for consistency with Figures 3 and 4

---

## Author Comment (AC1) · 4 Dec 2018

We thank the reviewer for their positive and useful comments.

[Figure]

Response to specific Comments:

1. A main comment would be that I find the title, and subsequent content a bit misleading in that it purports to provide monthly indices, whereas the indices in the paper are seasonal.

Thanks for highlighting something that was not clear in the manuscript. The indices that we provide are monthly indices calculated for seasonal windows (please, see the files uploaded in PANGAEA). To make this clear, we have modified Figures 3 and 4 to show the monthly indices instead of their seasonal averages. See our answer to point 2 below for further clarification.

2. A further methodological point is that it is not at all clear what is meant by "composites" throughout the paper, nor is it clear how these are constructed. I guess it is a combined index using different reanalyses, or do you mean combining monthly indices into seasonal indices? Exactly how these are combined should be made clear. I found this a bit confusing, but it should be stratightforward to clarify. Combination of time series from different reanalyses will involve splicing of some sort, and this should be explained clearly.

Indeed, the "composites" have been calculated averaging the reanalyses outputs for their overlapping periods. We have rephrased the methodology section 2.3 to clarify this point:

"Composite series of the NAO, EA and SCA patterns have been calculated for each 3-month season independently. Each individual month was given the average of the available EOF-based series with a confidence interval that corresponds to their standard deviation. The number of EOF-based series used for any given month is provided here along with the composite series. Since the EA and the SCA do not always correspond to the 2nd and 3rd EOF, respectively, a selection of what series to include in each composite based on their spatial patterns was done in advance (see Table 3 for a list of monthly EOFs included in each composite)."

We have also included an explanation sentence in the results section 3.2.1 to avoid confusions:

"Instead, since the correlations amongst datasets are very high (DJF: <0.9; MAM: >0.8; JJA: >0.6; SON: >0.9; Table S3), we have created robust composite series of each climate mode on the basis of their geographical representations as described in Table 3. This was done by averaging the overlapping EOF-based time-series that display either the NAO, EA or SCA (WA for MAM)."

Response to minor comments:

Page 1, Line 34: I would add the recent study by Hall and Hanna, 2018, IJOC, here, to broaden the scope of the literature. This paper also finds inconsistencies in EOFs 2 and 3 for summer.

We have included this reference.

Page 2, Line 6: Other nodes are used, such as Lisbon and Gibraltar, and this should be acknowledged and referenced here.

We agree with the referee and have re-written the sentence to include the main references using Lisbon and Gibraltar.

"The NAO is commonly described by an index calculated as the difference in normalized SLP over Iceland and the Azores (Cropper et al., 2015; Rogers, 1984), Lisbon (Hurrell and van Loon, 1997) or Gibraltar (Jones et al., 1997 ), but there are a number of robust alternatives to this classical definition of the NAO index such as Empirical Orthogonal Function analysis (EOF; Folland et al., 2009)."

Page 2, line 22: I would be more circumspect here. Although intuitively a positive EA should equate to positive SLP anomalies in line with SCA, the CPC index is based on the reverse of this, and a number of studies take this position (Woollings et al., 2010, QJRMS; Moore et al., 2011 QJRMS; Wulff et al., 2017, GRL; Hall and Hanna, 2018, IJOC among many) so it is incorrect to promote this view of the EA as the standard

one. It doesn't actually matter, the relationships are the same just inverted. It would be better to state: "Here we take the positive phase of the EA to be . . ... "

We have re-written this sentence as suggested:

"Here we use the positive phase of the EA as a strong centre of positive SLP anomalies offshore Ireland."

Page 2, line 29, Again it is appropriate to cite Hall and Hanna, 2018, IJOC

Done.

Page 3, lines 15-20. It is also worth noting that EOFS are statistical constructs and are not always associated with climate physics (Dommenget and Latif, 2002, J. Climate). Also some acknowledgement that the constructed EOFs are influenced by the region selected.

We agree with the reviewer's comment and have modified this paragraph accordingly.

". . . while EOF-based indices better capture the inter-annual variability in an area larger than the exact location of the centres of action (Folland et al., 2009), they are constrained by (i) the accuracy of the reanalysis products from which they are derived; (ii) the non-stationarity of the EOF pattern;(iii) the orthogonality imposed by the EOF technique; (iv) the fact that the constructed EOFs are influenced by the region selected; and (iv) having to repeat the analysis every time an update is required, which may change previously obtained time-series (Wang et al., 2014; Cropper et al., 2015). It is also worth noting that the EOFs are statistical constructs and are not always associated with climate physics (Dommenget and Latif, 2002)"

Pages 3-4 Data section. Were timeseries of station and gridded data assessed and corrected for any inhomogeneities which could arise through artificial means such as changing instrumentation, changes in density of records, etc?

Datasets were already tested for inhomogeneities by their corresponding sources:

Klein Tank et al. 2002 and MetEireann. We have noted this in the manuscript, where the instrumental data is introduced:

"Datasets were tested for inhomogeneities already by their sources (Table 1)."

For transparency, we have added another supplementary table with the details of all the meteorological stations that were considered, as well as a paragraph in the manuscript to explain the selection procedure:

"A set of meteorological stations were selected according to their proximity to the EA and SCA centres of action shown in our EOF analyses: Ireland for the EA and Norway for the SCA. Only one meteorological station with SLP measurements in Ireland could be used in this study: Valentia Observatory. On the other hand, five Norwegian stations with SLP data were located in the region of interest. The most suitable Norwegian station was further selected according to three criteria: i) length of the record, ii) continuity (i.e. the least missing data, the better) and iii) correlation with the EOF-based SCA time-series. Bergen Florida (Norway) was the station which better fulfilled these criteria. Details of all meteorological stations are available in Table S1."

Page 4, line 14. It is misleading to state that the common definition of a positive EA is positive SLP anomalies, in view of the comment and references above. Change to something like "our definitions"

We have followed the referee's suggestion.

"The polarities of the derived EOF time-series have been fixed to correspond to our definitions of the EA and the SCA (see section 1), which coincide with positive centres of action over the Atlantic and Scandinavia, respectively (Figs. 1 and S1-S4)."

Page 4, Line 32. How are the years of the moving windows defined, in reference to the window (start, end, centred-which is not possible with a 30 year window)?

Each time window is defined from i to i+30, where i is the oldest year of overlap between the time-series. We have added this text in the methodology section 2.4.

Page 6, Line 31: Are the composites monthly? They look seasonal to me. It is unclear from the text how the composites are produced. This needs to be explained clearly. This section is unclear, with confusing terminology about monthly time series when the figures show seasonal time series.

Following the first two general comments, we revised this section to avoid misunderstandings. Please see our answers to those comments above.

Page 8. Line 6. What is the 10-year filter? Is it a simple moving average, or some sort of Gaussian filter? The caption just says "bandpass" Can you be more specific?

We have included a brief explanation in the methodology section.

"Decadal variability of the time-series (Section 3.2.3) has been explored after filtering the time-series with a 2nd order low-pass Butterworth filter with a cut-off frequency of 1/10 (as implemented in the "butter" function of Matlab$^©$ R2018a)."

Page 8, lines 9-10 "...until a decrease towards a minimum starts in c. 1920" It is not clear what is meant by this as from the figure the minimum appears to be reached in 1920.

This point is addressed in the next comment, where we show the revised text.

Page 8, lines 11-14: I don't find these descriptions particularly convincing when looking at the figure

We agree with the reviewer that this text was not convincing, nor it was clear to the reader. The revised text reads as follows:

"Figures 3 and 4 show that most variability in EAcomp and SCAcomp is observed at inter-annual scales but some decadal variability is also evident in Figure 6. Overall, all 10-yr filtered indices fluctuate around the zero-line with no evident trend, except for one period when both series are persistently positive: during winter at the end of the 19th century (Fig. 6a). During this season, both indices show similar trends

between 1880 and 1920, when a decoupling occurs. In addition, the SCA experiences a large change of sign during the first three decades of the 20th century. Focusing on spring, we observe different patterns for both the EA and the WA with an EA absolute maximum at c. 1915 and two SCA minima at c.1930 and c.1960."

Page 8 line 15-16. I am highly sceptical about the reality of the first 20 years or so of the summer SCA figure, with its extreme maxima and minima. I think this is likely to be an artefact of data quality, See ESRL web pages https://www.esrl.noaa.gov/psd/data/gridded/20thC_ReanV2c/opportunities.html There is some evidence of this in Figure 6 panel b) as well.

We thank the reviewer for this information. To highlight the need to be cautious on this section of the data, we have highlighted it in all our figures with a grey band and have also modified the text to make this point clear:

"The extreme absolute minima at the start of the summer SCAcomp record (Fig. 4) seems to result from a low-pressure bias in marine records (Woodruff et al., 2005, Wallbrink et al., 2009) that has affected 20CRv2c fields such as the sea-level pressure from 1851 to c. 1865 (further information on this can be found here https://www.esrl.noaa.gov/psd/data/gridded/20thC_ReanV2c/opportunities). Since the 20CRv2c is the only reanalyses dataset covering that early period, we cannot provide an alternative. Instead, this period of low-confidence has been highlighted in all our figures with a grey band."

Responses to technical corrections

Page 1 line31: remove comma after "attention"

Done.

Page 5 line 29: should it be DJF: p>0.9?

Perhaps we do not understand the referee's suggestion but we believe the text in Page 5 line 29 is correct.

---

## Author Comment (AC2) · 4 Dec 2018

"Reconciling North Atlantic climate modes: Revised monthly indices for the East Atlantic and the Scandinavian patterns beyond the 20th century" by Laia Comas-Bru and Armand Hernández

We thank the reviewer for their comments.

Respose to general comments

1. Since the controlling mechanisms of the EA and the SCA are mentioned in the abstract, I would like to see a more careful description of the influence of these two patterns on climate variables and their physical meaning. Moreover, given the discrepancies across datasets and the lower percentage of variance explained by the patterns outside of the winter months, discussing their relevance would be appropriate.

We do not want to go into much detail on the influence of these two patterns on climate variables (this would be worth another manuscript!). However, we acknowledge that some indication on what these influences are would strengthen the manuscript. Therefore, to follow the referee's suggestion and to indicate the observed impacts on the main climate variables, we have now included a new section (section 3.4 Climate impact of the composite EA and SCA series) along with a new Figure 8. These show the correlations between the new monthly EA and SCA indices with precipitation amount and surface air temperature in Europe (where the correlations are expected to be more robust).

The new section reads as follows:

"Figure 8 illustrates the monthly correlation distribution maps between our composite-series (EAcomp and SCAcomp) versus surface air temperature and precipitation amount for the four seasons (DJF, MAM, JJA and SON) between 1901 and 2016 using the CRU-TS.4.01 dataset (Harris et al. 2014). The strongest correlations are found in winter, when these patterns are more prominent, and are consistent with previous studies (Moore et al, 2011; Comas-Bru and McDermott, 2014; Lim, 2015).

The only European regions for which the EA impacts on precipitation are strong and robust (i.e. on the same direction) throughout the year are the UK and Ireland. The predominantly weak correlations observed in other regions, far from the main centres of action, could arise from the low percentages of variability explained by each EOF pattern (<20% for EA; Table S22). Nevertheless, consistent patterns are observed in terms of precipitation amount across all seasons except in EAcomp/JJA, which also

shows an anomalous relationship with temperature. We interpret this to be caused by the northerly shift of the EA centre of action in JJA (i.e. between Scotland and Iceland instead of off-shore Ireland; see Table 3 and Figures S3 and S4), that hampers its influence on the western Mediterranean region, which in turn becomes wetter with positive EA modes. Regarding the impact of the SCA on precipitation, a similar pattern with negative correlations in northern Europe and predominantly positive correlations in the circum-Mediterranean region, is observed across seasons, albeit with different strengths.

We observe a strong seasonality on the impact of both climate modes on surface air temperature. Weak correlations are found for the all seasons except JJA for the EA with non-significant correlations across all Europe in SON. The opposite is observed for SCA, where the strongest impact on air temperature is shown in DJF (predominantly negative) and SON (predominantly positive). Due to the low variance explained by both climate modes, they are not expected to imprint a very strong signal on the climate and thus the extent to which these correlations would be reflected in the absolute precipitation and temperature values will primarily depend on the concomitant state of the NAO, the main driver of climate variability in the region (Hurrell and van Loon, 1997; Hurrell and Deser, 2010). In addition, the impact of these atmospheric modes on the climate is not robust throughout the year. For example, none of the datasets used in this study showed a SCA pattern within the three leading EOFs in spring.

Individual EOFs such as the EA and the SCA are statistical constructs that do not necessarily represent a physically independent phenomenon linked (i.e. correlated) to climate variables in a robust manner. Full characterisation of the regional atmospheric dynamics therefore requires multiple EOFs to be taken into account (Roundy, 2015). To thoroughly characterise the climate in the region, the impacts of the EA/SCA should be investigated in conjunction with the NAO (Moore et al., 2011; Comas-Bru and Mc-Dermott, 2014; Hall and Hanna, 2018) but this is outside the scope of this study. As far as we are aware, such investigation does not exist outside the winter months."

2. The main contribution of this paper is, in my opinion, the introduction of the new, instrumental indices that date back longer than previously available ones and that have been proven consistent across different datasets. The correlation of the winter SCA with the Bergen station data is remarkably good. For this reason, I think it would be interesting to know more about the instrumental data, and mention which other stations were considered and why they were discarded.

We agree with the reviewer's comments. A new paragraph introducing the selection criteria and the stations that we were considering has been added in section 2.1. In addition, we have listed the details of all these meteorological stations in a new supplementary table (S1).

"A set of meteorological stations were selected according to their proximity to the EA and SCA centres of action shown in our EOF analyses: Ireland for the EA and Norway for the SCA. Only one meteorological station with SLP measurements in Ireland could be used in this study: Valentia Observatory. On the other hand, five Norwegian stations with SLP data were located in the region of interest. The most suitable Norwegian station was further selected according to three criteria: i) length of the record, ii) continuity (i.e. the least missing data, the better) and iii) correlation with the EOF-based SCA time-series. Bergen Florida (Norway) was the station which better fulfilled these criteria. Details of all meteorological stations are available in Table S1."

Response to specific comments

Page 1, Line 25: "The spatial structure of climate changes..." I would suggest talking about regional climate variability rather than 'climate changes', given the data span no longer than two hundred years.

Done: "The spatial structure of regional climate variability follows recurrent patterns often referred to as modes of climate variability or teleconnections, which provide a simplified description of the climate system (Trenberth and Jones, 2007)."

Page 2, Line 21: A strong centre of positive SLP anomalies is said here to be associated with above-average temperature and wetter conditions in Northern Europe. Please review this. Is it possible that those effects correspond to the positive phase of the reverse EA index, as used in other publications? (i.e., Moore et al., 2011)

Indeed! We have rephrased the sentence to match the direction of our EA index.

"Here we use the positive phase of the EA as a strong centre of positive SLP anomalies offshore Ireland. This is associated with below-average surface temperatures in Southern Europe, drier conditions over Western Europe and wetter conditions across much of Eastern Europe and the Norwegian coast (Moore et al., 2011; Rodríguez-Puebla and Nieto, 2010)."

Page 3, Line 18: The orthogonality imposed by the EOF technique should be one of the constraints listed

Agreed. We have now added this as one of the constrains:

". . . while EOF-based indices better capture the inter-annual variability in an area larger than the exact location of the centres of action (Folland et al., 2009), they are constrained by (i) the accuracy of the reanalysis products from which they are derived; (ii) the non-stationarity of the EOF pattern;(iii) the orthogonality imposed by the EOF technique; (iv) the fact that the constructed EOFs are influenced by the region selected; and (iv) having to repeat the analysis every time an update is required, which may change previously obtained time-series (Wang et al., 2014; Cropper et al., 2015). It is also worth noting that the EOFs are statistical constructs and are not always associated with climate physics (Dommenget and Latif, 2002)"

Page 5, Line 6: Looking at Table 3, no discrepancies are observed across datasets for EOF 3 during MAM months

The referee is right. We have removed this from the text.

"For example, while the geographical patterns are very stable across datasets during

winter (Table 3), some discrepancies are observed during summer (JJA; see EOF2 or EOF3)."

Page 5, Line 21: "Because of this spatial pattern, ..." I suggest rephrasing the sentence since the meaning is not entirely clear to me.

We have rephrased the sentence and hope it is now clearer:

"Due to the spatial extent of the winter's EOF3 positive centre of anomalies covering from Scandinavia to SW Ireland, ValSLP (purple dot in Figure 1 and S1-S4) is unsurprisingly correlated with all winter EOF3s (0.5<<0.6; Table 4)."

Page 6, Line 27: I think it is erroneous to assume that the datasets have to capture the climate modes, since the variability is in the SLP data themselves. Comparing different reanalysis datasets is out of the scope of this paper, but an indication of known quality issues that might account for the differences would be suitable

Following reviewer's 1 comment, we have now added some information on the anomalous SLP observed in the 20CRv2c dataset for the period 1951-1965. This has also been highlighted in all figures with a grey box.

In order to address the reviewer's concern, we now direct the reader to Fujiwara et al., 2017, which provides an extensive review of the quality of all the reanalyses products used in this study. We've also added a short paragraph highlighting the main differences of each of the reanalyses datasets used in this study.

"Five reanalyses datasets have been used in this study (Table 2). ERA-40 (Uppala et al., 2005) is a conventional-input reanalysis used in many studies that require long-term atmospheric data. ERA-Interim (Dee et al., 2011) improves ERA-40 in that it assimilates a more complete set of observations and therefore it achieves more realistic representations of the hydrologic cycle and the stratospheric circulation relative to ERA-40, as well as it improves the consistency of the reanalysis products over time. ERA-20C (Poli et al., 2016) directly assimilates surface pressure and surface wind ob-

[Figure]

servations, enabling it to extend back in time to cover the entire 20th century. 20CRv2c (Compo et al., 2011) is also a surface-input reanalysis with a different assimilation procedure than that of ERA-20C. The main limitation of 20CRv2c is that it does not correct for biases in surface pressure observations from ships and buoys, which results in the anomalous SLP observed for the period 1850-1865. Finally, the NCEP/NCAR (Kalnay et al., 1996) was the first modern reanalysis of extended temporal coverage (1948 to present) and it is still widely used. For an extensive review on the quality of these datasets, the reader is referred to Fujiwara et al., 2017."

Figure 2: many data are shown on the same plot, so that it is a bit difficult to visually recognise the agreement between the series. Also, could the x axis show dates rather than number of months?

Done.

Figure 6: I suggest including DJF, MAM, JJA and SON instead of a, b, c and d on the top left corner of each panel, both for clarity and for consistency with Figures 3 and 4

Done.